

# A molecular dynamics simulation study of the ACE2 receptor with screened natural inhibitors to identify novel drug candidate against COVID-19

Neha Srivastava[1], Prekshi Garg[2], Prachi Srivastava[2] and
Prahlad Kishore Seth[3]

[1] Bioinformatics Centre, Biotech Park, Lucknow, Uttar Pradesh, India
[2] Institute of Biotechnology, AMITY University, Lucknow, Uttar Pradesh, India
[3] NASI Senior Scientist Platinum Jubilee Fellow, Biotech Park, Lucknow, Uttar Pradesh, India

## ABSTRACT

**Background & Objectives.** The massive outbreak of Novel Severe Acute Respiratory Syndrome Coronavirus (SARS-CoV-2) has turned out to be a serious global health issue worldwide. Currently, no drugs or vaccines are available for the treatment of COVID-19. The current computational study was attempted to identify a novel therapeutic inhibitor against novel SARS-CoV-2 using in silico drug discovery pipeline.

**Methods.** In the present study, the human angiotensin-converting enzyme 2 (ACE2) receptor was the target for the designing of drugs against the deadly virus. The 3D structure of the receptor was modeled & validated using a Swiss-model, Procheck & Errat server. A molecular docking study was performed between a group of natural & synthetic compounds having proven anti-viral activity with ACE2 receptor using Autodock tool 1.5.6. The molecular dynamics simulation study was performed using Desmond *v 12* to evaluate the stability and interaction of the ACE2 receptor with a ligand.

**Results.** Based on the lowest binding energy, confirmation, and H-bond interaction, cinnamic acid (−5.20 kcal/mol), thymoquinone (−4.71 kcal/mol), and andrographolide (Kalmegh) (−4.00 kcal/mol) were screened out showing strong binding affinity to the active site of ACE2 receptor. MD simulations suggest that cinnamic acid, thymoquinone, and andrographolide (Kalmegh) could efficiently activate the biological pathway without changing the conformation in the binding site of the ACE2 receptor. The bioactivity and drug-likeness properties of compounds show their better pharmacological property and safer to use.

**Interpretation & Conclusions.** The study concludes the high potential of cinnamic acid, thymoquinone, and andrographolide against the SARS-CoV-2 ACE2 receptor protein. Thus, the molecular docking and MD simulation study will aid in understanding the molecular interaction between ligand and receptor binding site, thereby leading to novel therapeutic intervention.

Corresponding author
Prachi Srivastava,
psrivastava@amity.edu

## INTRODUCTION

Novel Severe Acute Respiratory Syndrome Coronavirus (SARS-CoV-2) is a unique virus with an unusually large positive-sense RNA genome ranging from o 32 kb in length (*Lu et al., 2020*) and a special replication strategy. It is a member of the largest group of viruses, Nidovirales (*Naqvi et al., 2020*). It is enveloped and characterized by club-like spikes that project out from the surface. The receptor-binding domain of spike protein binds with the host cell and makes the virus capable of causing a variety of diseases in the host (*Huang et al., 2020*). It causes enteritis in cows and pigs and upper respiratory disease in birds. It is now proven to cause potentially lethal respiratory infections in humans. Before the outburst of the coronavirus, it was thought that these viruses cause only mild, respiratory infections in humans but after the outburst, lethal side effects of the virus have become the major global concern. These viruses are known to cause more severe diseases in neonates, the elderly, and individuals with underlying illnesses (*Perlman, 2016*).

In December 2019, Wuhan, China became the center of the outburst of the novel SARS-CoV-2 virus which raised intense and urgent concern internationally. A similar outburst of the SARS-CoV virus was also experienced in 2003 in Guangdong, China. The number of cases at that time increased substantially and spread globally (*Zhong et al., 2003*). In 2003, the virus-infected approximately 8,000 people and caused 774 deaths worldwide (*Sarkar & Saha, 2020*). The major reason for the spread of SARS-CoV in 2003 was the ability of the virus to transmit rapidly among people. Also, the insufficient preparedness and implementation of control measures led to the further spread of the virus. Learning from the past experiences, isolation of people suspected to have the disease, close monitoring of their contacts, epidemiological and clinical data collection from patients, and development of diagnostic as well as treatment procedures are some of the widely accepted measures adopted to control the spread of novel coronavirus (*Wang et al., 2020*). The 2019 outbreak of novel SARS-CoV-2 has infected more than 66,818,411 people with 1,428,870 deaths worldwide as of November 2020. The control of the spread of the virus and also an adequate plan of action and treatment of the virus is the major need of the hour. Therefore, researchers are now trying to find a drug that can considerably treat the effects of the virus. In the present study, we aim to identify compounds that can be used as prophylactics against novel SARS-CoV-2. Coronavirus contains a genome of approximately 30 kb and consists of a 5′ cap structure and 3′ poly(A) tail that allows it to act as mRNA for the translation of the replicase polyproteins. At the beginning of each structural or accessory gene, there is the presence of transcriptional regulatory sequences (TRSs) which are essential for the expression of each of these genes. SARS-CoV-2 is known to bind with angiotensin-converting enzyme 2 (ACE2) as its receptor to gain entry into the human cells (*Li et al., 2003*). Various non-structural proteins assemble into the replicase-transcriptase complex of the viral genome to provide an environment suitable for the synthesis of RNA and also contain enzyme domains and functions which are important for the replication of RNA. Various non-structural proteins (nsp) play an important role in spreading and activating the virus inside the host. Nsp1 promotes cellular mRNA degradation and blocks host cell translation, which ultimately results in the

**Table 1 Comparison between SARS-CoV, MERS-CoV, and SARS-CoV-2.**

| Characteristic | SARS-CoV | MERS-CoV | SARS-CoV-2 |
|---|---|---|---|
| Year in which the first case was reported | 2002 | 2012 | 2019 |
| Country/region where the first case was reported | China | Middle East | China |
| The primary mode of transmission | Droplets, aerosol, and contact | Droplets, aerosol, and contact | Droplets, aerosol, and contact |
| Incubation period | 2–7 days | 2–14 days | 2–14 days |
| Host receptor | ACE2 | DPP4 | ACE2 |
| Case fatality rate | Approximately 15% | 34.4% | 1–3% |

blockage of the innate immune response (*Huang et al., 2011*; *Tanaka et al., 2012*). Nsp2 binds to prohibition proteins and further enhances the spread of the virus in the body (*Cornillez-Ty et al., 2009*). Nsp3 is a large, multi-domain transmembrane protein consisting of Ubl1 and Ac domains that interact with N protein, ADRP activity that promotes cytokine expression, and PLPro/Deubiquitinase domain that cleaves viral polyprotein and blocks host innate immune response (*Chatterjee et al., 2009*; *Frieman et al., 2009*; *Serrano et al., 2009*). Nsp4 and nsp6 are potential transmembrane scaffold proteins, important for the proper structure of DMVs (*Gadlage et al., 2010*; *Oostra et al., 2008*). Nsp5 cleaves viral proteins, nsp7, and nsp8 act as a clamp for RNA polymerase (*Imbert et al., 2006*) nsp9 is an RNA binding protein, and nsp13 acts as RNA helicase for the viral genome. Nsp10 acts as a cofactor for nsp14 (an important protein for proofreading of the viral genome (*Eckerle et al., 2010*)) and nsp16 (shields viral RNA from MDA5 recognition (*Zust et al., 2011*)) and stimulates ExoN and 2-0-MT activity (*Bouvet et al., 2010*; *Decroly et al., 2011*).

Novel SARS-CoV-2 is very similar to SARS-CoV (*Andersen et al., 2020*; *Lu et al., 2020*; *Zhu et al., 2020*). Both these viruses belong to the betacoronavirus genus (lineage B) (*Chan et al., 2020*; *Letko, Marzi & Munster, 2020*). The detailed comparison of SARS-CoV, MERS-CoV, and SARS-CoV-2 is depicted in Table 1 (*Abdelrahman, Li & Wang, 2020*).

The entry of the virus into the cell is one of the most primary and essential steps for the spread of disease caused by novel SARS-CoV-2. This virus encodes a surface glycoprotein known as spike (*Li, 2016*). The receptor-binding domain (RBD) of the spike protein is responsible for binding with the host cell receptor and mediates the entry of the virus into the cell. The virus enters the host cell due to the release of spike fusion peptide formed as a result of cleavage of spike protein by the host protease (*Simmons et al., 2013*; *Bertram et al., 2011*). The similarity in the receptor-binding domain of both SARS-CoV-2, suggests that both these viruses share a common receptor, ACE2. Angiotensin-converting enzyme 2 (ACE2) is primarily found in alveolar epithelial cells (*Zhao, 2020*) and is responsible for reducing the surface tension of these cells, thereby preventing the collapse of alveoli. Therefore, ACE2 is a very important protein to ensure proper gas exchange through the lungs. Any injury caused in such cells will lead to severe lung injury, which was commonly observed in COVID-19 patients. The binding of the spike protein of SARS-CoV-2 with ACE2 of host cell leads to endocytosis of virus and loss of ACE2

function, therefore due to this binding, the cell loses its main component which was responsible for protecting the lungs from any injury. This makes the host vulnerable to viral infection. The receptor-binding domain of the spike glycoprotein binds to the tip of subdomain 1 of ACE2. The viral membrane fuses with the host cell and activates the cell (*Wrapp et al., 2020*; *Song et al., 2018*; *Li et al., 2005*). After fusion, viral RNA is released into the cytoplasm that establishes infection. Thus, targeting the ACE2 receptor of host cells can block the entry of the virus into the cell, thereby protecting the host from viral infection and pandemic disease COVID-19.

Ever since the outburst of the pandemic, researches are going on worldwide, to explore different aspects of SARS-CoV-2 and the mechanism of infection. From repurposing drugs to finding new drugs and vaccine candidates, the studies on SARS-CoV-2 have come a long way. Some of the recent studies explore new immunogenic dimensions of the pandemic. In one of the studies, the researchers took the ontological approach to identify targets, both in the virus and the host, that can aid in searching for effective vaccines and drugs against COVID-19 (*Jayachandran et al., 2020*). In another study, researchers conducted molecular modeling intending to reveal new druggable binding sites in the spike protein of SARS-CoV-2. In their study, they disclosed 8 novel druggable binding sites of spike protein (*Ugur Marion & Marion, 2020*). In one of the studies, the scientists tested the potential of flavonols to act as antiviral drugs by targeting spike protein, SARS-CoV-2 proteases, RNA-dependent RNA polymerase, and ACE2 receptor (*Mouffouk et al., 2020*).

Therefore, there is a need for more effective and rational approaches that open the door to the development of more effective new therapeutics targets and drug candidates against Covid-19. In this regard, computational approaches play a significant role in the process of rapid drug discovery and development process with less time and cost-effective manner. Besides this, herbal medication shows a hugely significant impact on human health (*Pan et al., 2014*). Nearly about 70–80% of the world population relies on herbal drugs due to their great compatibility with lower/zero side-effects on human health. The bio-active constituents present in the plants possess medicinal properties is used by various pharmaceutical and R&D industries for the development of drug due to their safer and better use (*Ekor, 2014*). In the current insilco study, drug discovery approaches are implemented to identify novel therapeutic inhibitors against novel SARS-CoV-2 targeting the human Angiotensin-converting enzyme 2 (ACE2) receptor.

# METHODOLOGY

## Protein structure prediction and validation

The protein sequence of the ACE2 receptor was retrieved in FASTA format from the NCBI public database. For the identification of similar sequences, the BLAST program was performed for the search of structurally similar sequences with a protein databank database (PDB) (*Gupta et al., 2013*). The 3D structure of the ACE2 receptor was modeled using online modeling server SWISS-Model (*Waterhouse et al., 2018*) based on the BLAST parameter. Further, the energy minimization of the modeled structure was done using Chimera 1.10.1 tool (*Goddard, Huang & Ferrin, 2005*). The 3D structure was subjected to validation using Procheck and Errat Plot. The Ramachandran plot (*Spencer et al., 2007*)

**Table 2 List of natural compounds.**

| S. No. | Compound name | Plant derived | Mol. wt (g/mol) | H-bond donor | H-bond acceptor |
|--------|---------------|---------------|-----------------|--------------|-----------------|
| 1. | Choline | *Withania Somnifera* | 104.17 | 1 | 2 |
| 2. | Harmine | *Passiflora Incarnata* | 182.23 | 1 | 2 |
| 3. | Cinnamaldehyde | *Cinnamonum Cassia* | 132.16 | 0 | 1 |
| 4. | Cinnamic Acid | *Bromeliad* | 148.16 | 1 | 2 |
| 5. | Coumarins | *Dypsis Lutescens* | 146.15 | 0 | 2 |
| 6. | Ursolic Acid | *Ocimum Basilicum* | 456.7 | 2 | 3 |
| 7. | Chlorogenic Acid | *Daucus Carota* | 354.31 | 6 | 9 |
| 8. | Assafoetidnol B | *Ferula Assafoetida* | 456.54 | 2 | 7 |
| 9. | Glucobrassicin | *Brassica Oleracea* | 448.48 | 6 | 11 |
| 10. | Linamarin | *Phaseolus Lunatus* | 247.25 | 4 | 7 |
| 11. | Luteolin | *Thymus Vulgaris* | 286.24 | 4 | 6 |
| 12. | Andrographolide(Kalmegh) | *Andrographis Paniculata* | 350.4 | 3 | 5 |
| 13. | Catechin | *Camellia Sinensis* | 458.4 | 15 | 12 |
| 14. | Citral | *Melissa Officinalis* | 152.24 | 0 | 1 |
| 15. | Hydroxychloroquine | *Synthetic* | 335.9 | 2 | 4 |
| 16. | Linalool | *Lavandula Angustifolia* | 154.25 | 1 | 1 |
| 17. | Nicotine | *Nicotiana Tabocum* | 162.23 | 0 | 2 |
| 18. | Chloroquine | *Helianthus Annus* | 319.9 | 1 | 3 |
| 19. | Allium | *Allium Sativum* | 150.3 | 0 | 0 |
| 20. | Thymoquinone | *Nigella Sativa* | 164.2 | 0 | 2 |

evaluates any steric clashes and structure reliability whereas the errat plot measures the overall error frequency rate in modeled structure.

## Ligand identification

The Thirty natural & synthetic compounds having proven in-vitro & in-vivo anti-viral activity was used in the study based on various literature reviews are depicted in Tables 2 and 3 (*Manoj Kumar et al., 2014*). The available 2D & 3D structures of compounds were retrieved from the PubChem database. Molinspiration chemoinformatics software was used to predict the Molecular and bioactivity analysis as well as drug-likeness properties of these compounds.

## Molecular docking study

A molecular docking study was performed between the ACE2 receptor and 30 natural & synthetic compounds using the Autodock 1.5.6 tool to predict the best binding score, affinity, and confirmation (*Yue et al., 2016*; *Yue et al., 2017*). AutoDock is an automated suite of protein-ligand docking tools used to predict the protein interactions with small molecules such as drug molecules and substrate. It analyzes the interaction of ligand molecules at the specified target site of the protein. AutoDock uses AMBER forcefield and linear regression to predict the free binding energies (*Ravi & Kannabiran, 2016*).
The lowest binding energy was the criteria undertaken to select the best protein-ligand

| Table 3 List of synthetic compounds. | | | | |
|---|---|---|---|---|
| S. No. | Compound name | Mol. wt (g/mol) | H-bond donor | H-bond acceptor |
| 1. | Amprenavir | 505.64 | 1 | 4 |
| 2. | Acyclovir | 225.21 | 4 | 8 |
| 3. | Umifenovir | 477.42 | 1 | 5 |
| 4. | Combivir | 505.5 | 5 | 18 |
| 5. | Tamiflu | 312.41 | 3 | 6 |
| 6. | Zanamivir | 332.31 | 9 | 11 |
| 7. | Cidofovir | 279.19 | 5 | 9 |
| 8. | Peramivir | 328.41 | 7 | 8 |

complex. Calculation of binding energy was performed using a semi-empirical free energy force field with charge-based desolvation and grid-based docking. The force field was selected based on a comprehensive thermodynamic model that allows the incorporation of intermolecular energies into the predicted binding energy (*Nakamura et al., 2010*). Cygwin platform was used to run docking files. Discovery Studio 4.5 was used to visualize and analyze protein-ligand interaction and confirmation.

## Molecular dynamic simulation study

The molecular dynamic simulation study was performed to examine the conformational changes in the protein that occurred due to the ligand-binding site and to evaluate the effect of these changes over the protein-ligand complex (*Yue et al., 2017*; *Okimoto et al., 2009*). To evaluate the stability and interaction of the ACE2 receptor with ligand simulation study was performed using Desmond *v12* Schrödinger software package at 50 ns time period (*Wright et al., 2020*). The complex was placed in protein preparation wizard for optimization, analysis, and refinement of docking complex, system builder menu set up membrane model POPC (Palmitoyl Oleoyl Phasphatidyl chlorine). Water molecules were added to the docking complex of the ACE2 receptor with a simple point charge (SPC) water model. System builder was build using counter ions, shake algorithm used to constrain the geometry of water molecules, and heavy atom bond lengths with hydrogen, electrostatic interaction applied using Particle Mesh Ewald (PME) method and Orthorhombic were used as periodic boundary conditions (PBC). The energy was minimized with 5,000 steps maximum iteration using the steepest descent algorithm (SD) and 1,000 steps using the conjugate-gradient algorithm (CG) with convergence threshold 50 e. Dynamic was performed with 50 ns, during the simulation the length of bond involving hydrogen was constrained using NPT ensemble, without restraints, for a simulation time of 1.2 picoseconds (ps) (temperature 300 K) was performed to relax the system.

## RESULT

### 3D structure prediction & validation

The total length of the ACE2 receptor in humans is 805 amino acids. The 3D model of protein was modeled using the online modeling server SWISS-Model. Based on the lowest

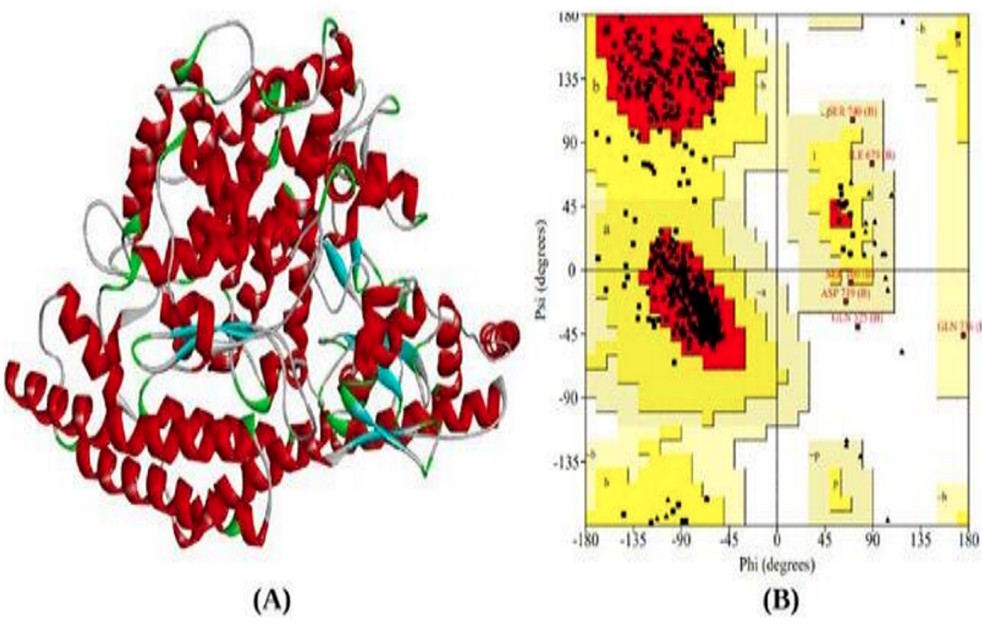

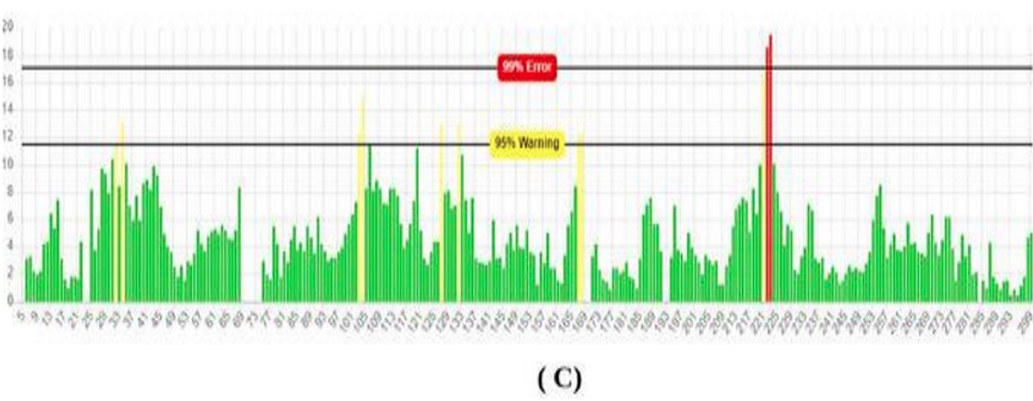

**Figure 1 Protein modeled structure.** Structure prediction & validation of ACE2 Receptor. (A) Modeled 3D structure of ACE2 receptor. (B) Ramachandran plot of ACE2 receptor generated by procheck server. (C) Error frequency plot generated by ERRAT server.

z-score and valid q-mean score the best model was predicted. The modeled 3D structure was subjected to validation using Procheck and Errat Plot. The Ramachandran plot was used to evaluate the position of amino acid residues in the allowed and disallowed region and the overall stereochemical property of the protein structure. Errat plot was used to calculate the overall error frequency rate. The Ramachandran plot showing 90.9% in the favored region; indicate good structure stability and reliability. The errat plot showing quality factor 96.08% reveals non-bonded interaction with different atoms in the modeled structure which indicates good quality of the model (Fig. 1).

## Molecular docking study

Molecular docking study is known to be the most reliable method to analyze and predict the best fit protein-ligand confirmation (*Liu et al., 2015*; *Yue et al., 2016*). A molecular

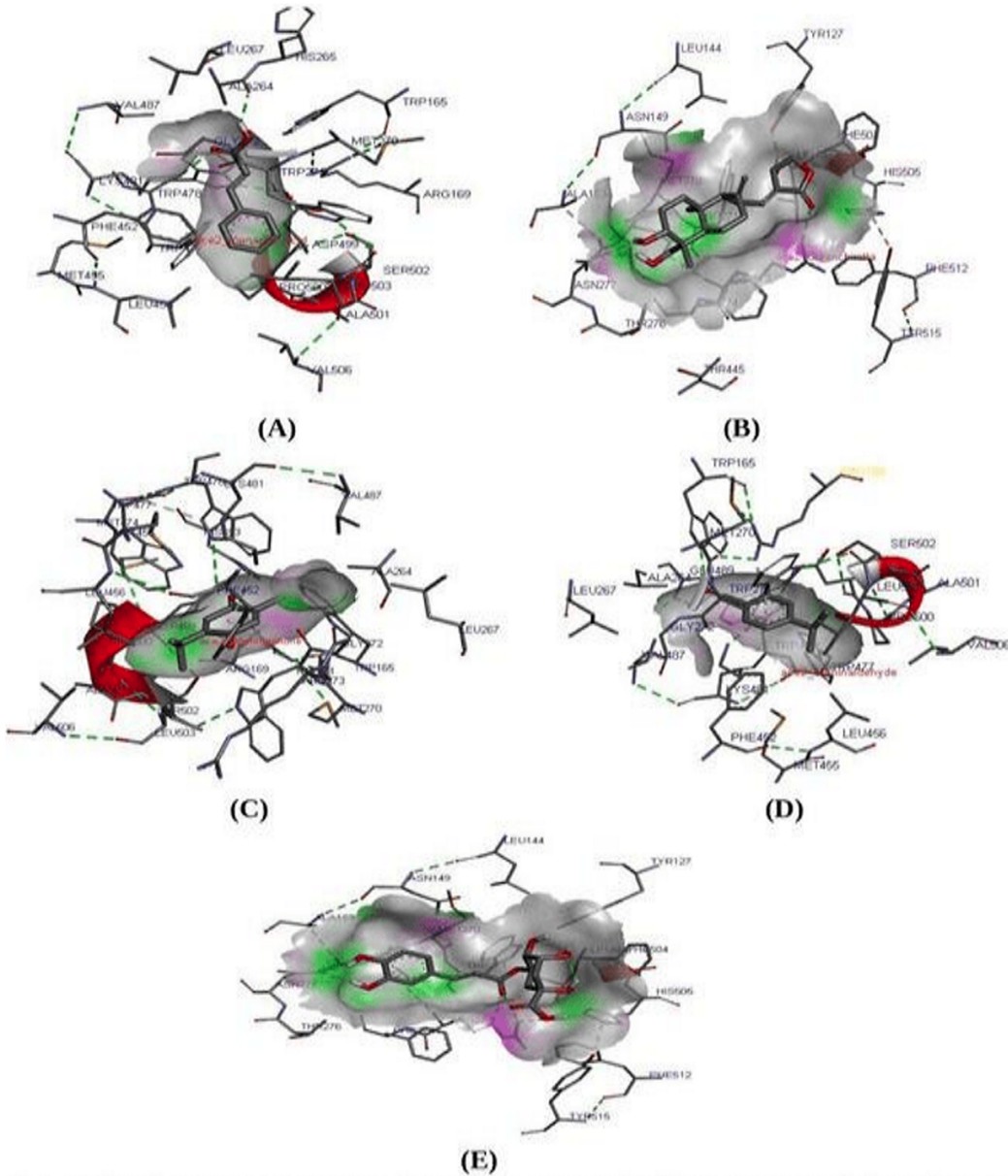

**Figure 2 Molecular interaction study.** Molecular interaction study of ACE2 receptor with (A) Cinnamic acid (B) Thymoquinone (C) Andrographolide (D) Cholengenic acid (E) Cuminaldehyde. Here in the fig, the green dot indicates the H-bond interaction between protein-ligand.

docking study was performed between ACE2 receptor and thirty compounds using Autodock 1.5.6 tools to obtain a higher stable protein-ligand complex (Table 4). Based on the lowest binding energy ($\Delta G_b$, kcal/Mol), confirmation, hydrogen bond interaction, the three compounds namely Cinnamic acid (−5.20 kcal/mol), Thymoquinone (−4.71 kcal/mol), and Andrographolide (Kalmegh) (−4.00 kcal/mol) were screened out showing strong binding affinity to the active site of ACE2 receptor (Fig. 2 & Table 5).

**Table 4 Molecular docking results.**

| S. No. | Compound name | Binding energy (Kcal/Mol) |
|---|---|---|
| 1. | Choline | −3.99 Kcal/Mol |
| 2. | Harmine | −3.71 Kcal/Mol |
| 3. | Cuminaldehyde | −4.00 Kcal/Mol |
| 4. | Cinnamic Acid | −5.20 Kcal/Mol |
| 5. | Curcumrin | −2.42 Kcal/Mol |
| 6. | Ursolic Acid | −3.43 Kcal/Mol |
| 7. | Chlorogenic Acid | −4.20 Kcal/Mol |
| 8. | Assafoetidnol B | −3.82 Kcal/Mol |
| 9. | Glucobrassicin | −3.65 Kcal/Mol |
| 10. | Linamarin | −2.32 Kcal/Mol |
| 11. | Luteolin | −4.19 Kcal/Mol |
| 12. | Amprenavir | −0.84 Kcal/Mol |
| 13. | Acyclovir | −3.24 Kcal/Mol |
| 14. | Umifenovir | −2.30 Kcal/Mol |
| 15. | Combivir | −1.23 Kcal/Mol |
| 16. | Tamiflu | −3.14 Kcal/Mol |
| 17. | Zanamivir | −2.00 Kcal/Mol |
| 18. | Thymoquinone | −4.71 Kcal/Mol |
| 19. | Cidofovir | −1.78 Kcal/Mol |
| 20. | Peramivir | −3.84 Kcal/Mol |
| 21. | Andrographolide(Kalmegh) | −4.63 Kcal/Mol |
| 22. | Greentea | −3.25 Kcal/Mol |
| 23. | Citral | −2.82 Kcal/Mol |
| 24. | Hydroxychloroquine | −2.70 Kcal/Mol |
| 25. | Linalool | +1.49 Kcal/Mol |
| 26. | Nicotine | −1.27 Kcal/Mol |
| 27. | Chloroquine | −2.52 Kcal/Mol |
| 28. | Allium | + 0.58 Kcal/Mol |

**Table 5 Best five docking results.**

| S. No. | Compound name | Binding energy (Kcal/Mol) | H-bond interaction |
|---|---|---|---|
| 1. | Cinnamic acid | −5.20 kcal/Mol | ALA-264,LYS-481, LEU-503 |
| 2. | Thymoquinone | −4.71 kcal/Mol | LEU-103, TRP-165, ARG-169 |
| 3. | Andrographolide (Kalmegh) | −4.63 Kcal/Mol | ARG-273, TYR-127, ASN-277, ASN-277 |
| 4. | Cholengenic acid | −4.20 kcal/Mol | ARG-273, ARG-273 |
| 5. | Cuminaldehyde | −4.00 Kcal/Mol | TRP-165 |

Thus these compounds may be considered as a suitable therapeutic inhibitor against SARS-CoV-2. Further, all three dock complexes were subjected to a molecular dynamics simulation study to check the stability of protein-ligand complexes.

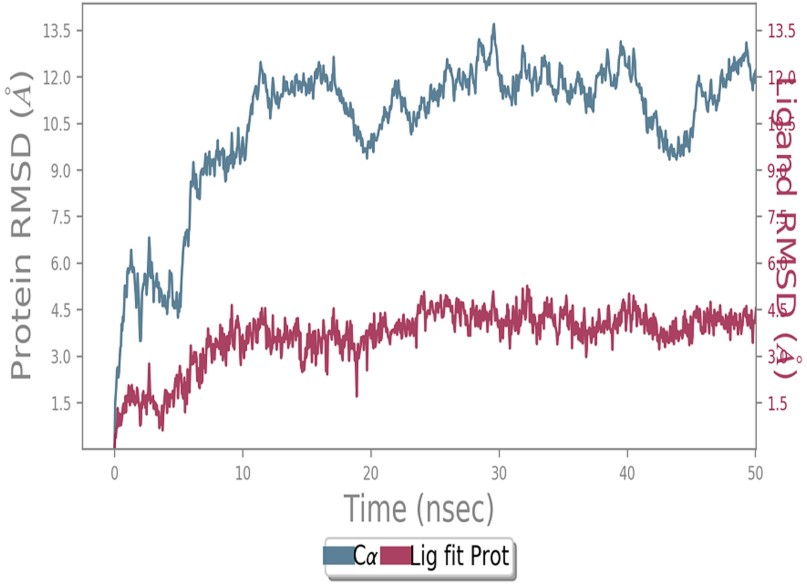

**Figure 3 Molecular dynamic simulation of ACE2-Cinnamic acid.** Root mean square deviation (RMSD) plot for ACE2—cinnamic acid complex during 50 ns of molecular dynamics simulation. ACE2 is shown in the red color and cinnamic acid in the blue color.

## Molecular dynamic simulation study

MD simulation study was performed to check the stability of the ACE2 receptor with cinnamic acid, thymoquinone, and green chireta using Desmond *v12* based on docking results and h-bond interaction study. To study the stabilities, root means square deviation (RMSD) was calculated with respect to the initial structures along the 50 ns (ns) each trajectory with average RMSD (Root mean square deviation) 12.0A – 4.5A for ACE2-cinnamic acid complex (Fig. 3), 12.0 A–6.0 A for ACE2-thymoquinone complex (Fig. 4), and 12.0 A–6.0 A for ACE2-Andrographolide (Kalmegh) (Fig. 5). In summation, complex stability also proved the validity of the docking results. There is no effect of temperature and pressure on the conformation of the structure.

## DISCUSSION & CONCLUSION

The causative agent of COVID-19 respiratory disease, novel coronavirus SARS-CoV-2 pandemics is occurring as a life-threatening disease infecting more than 66,818,411 people worldwide. Nearly about 1,428,870, total deaths estimated so far. At current, there are no available antiviral drugs or vaccines with proven efficacy for the prevention and treatment of the diseases. Therefore, a more effective and rational approach is needed that will ultimately lead to new therapeutic approaches. In this regard, the computational approach holds promising trends in the drug discovery & development process with reducing cost and time. In the current insilico study, the angiotensin-converting enzyme 2 (ACE2) receptor was taken as the target protein as various literature studies suggested that ACE2 as a functional SARS-CoV-2 receptor required for host cell entry and subsequent viral replication (*Kuba et al., 2005*). The 3D structure of the ACE2 receptor
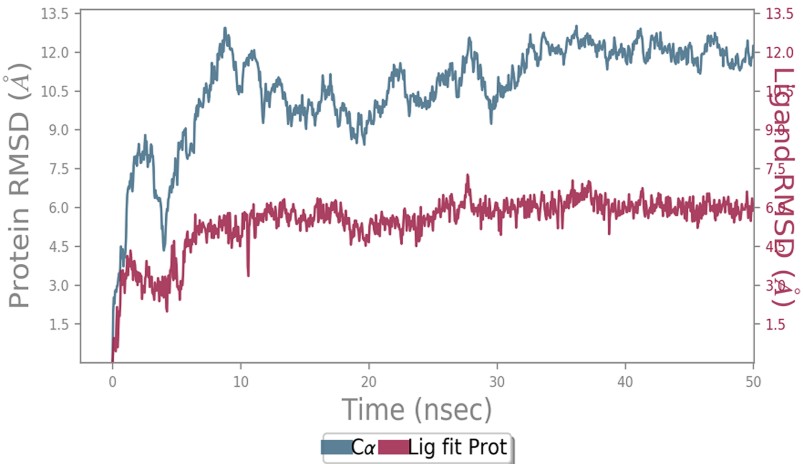

**Figure 4  Molecular dynamic simulation study of ACE2-Thymoquinone.** Root mean square deviation (RMSD) plot for ACE2—Thymoquinone complex during 50 ns of molecular dynamics simulation. ACE2 is shown in the red color and Thymoquinone in the blue color.

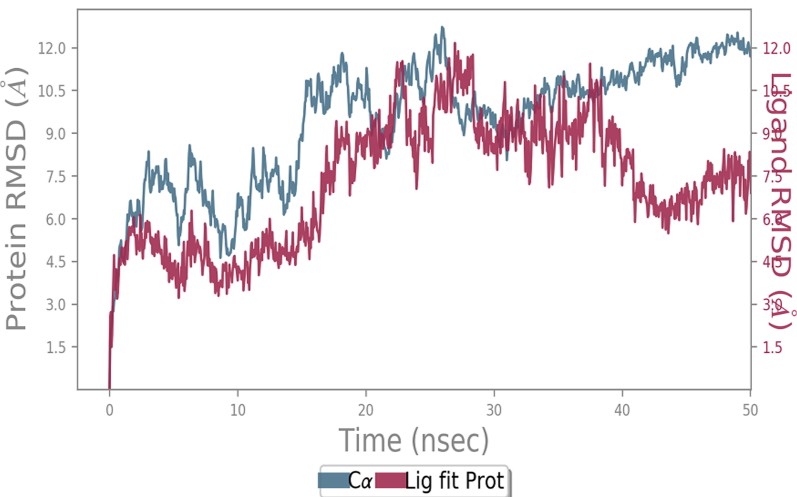

**Figure 5  Molecular dynamic simulation study of ACE2-Andrographolide.** Root mean square deviation (RMSD) plot for ACE2—Andrographolide (Kalmegh) complex during the 50 ns of molecular dynamics simulation. ACE2 is shown in the red color and Andrographolide (Kalmegh) in the blue color.

was modeled and validated. A docking study of filtered natural & synthetic compounds was performed with the ACE2 receptor. Based on the lowest binding energy, docking confirmation, and H-bond and pi–pi interaction Cinnamic acid (−5.20 kcal/mol), Thymoquinone (−4.71 kcal/mol), and Andrographolide (Kalmegh) (−4.00 kcal/mol) showing strong binding at the active site of ACE2 receptor. Further, The MD simulation result reveals that cinnamic acid, thymoquinone, and andrographolide could efficiently activate the biological pathway without changing the conformation in the binding site of

the ACE2 receptor. The bioactivity and drug-likeness properties of compounds show their better pharmacological property and safe to use.

Cinnamic acid derived from the plant *Bromeliad*, have a virucidal activity, involved in the inhibition of the viral replication cycle including the formation of the viral envelope and cell membrane thereby affecting the whole epidemiology of the viral protein. It is known to be an excellent phytochemical that have cytopathic activities against all viral infections. The study reported (*Orhan & Sezer Senol Deniz, 2020*) the potential of plant extracts and natural compounds against SARs CoV-2 and other deadly viruses showed the suppressive effect of *trans*-cinnamic acid (IC50 = 3.0 ± 0.18 μg/ml, SI = 7.4) on Covid-19 infection. The molecular docking study conducted on Vero cell against SARS CoV-2 viral infection targeting Mpro protein showed the highest activity with cinnamic acid (*Santos et al., 2020*). Thymoquinone derived from the plant *Nigella sativa*, significantly enhances immune responsiveness and suppresses pathogenicity of many viruses. The study suggested that it blocked the replication of SARS-CoV spike (S) protein, which plays a major role in host cell viral attachment to receptor angiotensin-converting enzyme 2 (ACE2). The study reports the antinociceptive effects of thymoquinone produce via indirect activation of supraspinal μ1- and κ-opioid receptors (*Abdel-Fattah, Matsumoto & Watanabe, 2000*). In reference to this hemorphins, opoid active peptide also block the effect of ACE (*Lantz et al., 1991*). Thus these findings suggest a similar inhibitory molecule for both opoid receptor and ACE2. Therefore, thymoquinone might be proved as a novel inhibitory molecule for ACE2 in SARS-CoV-2. In an in-Silico study (*Mohideen, 2021*) targeting E protein of SARs CoV-2 shows good binding affinity and confirmation (BE = −9.01 kcal/mol) with thymoquinone using Argus Lab 4.0. Further, their comparative study with patch dock also showed better affinity (BE = −32.03 kcal/mol). Thus these findings suggest the effectiveness of thymoquinone against SARsCoV-2 targets. In the study on a natural compound having involvement in ER stress against HSPA5 substrate-binding domain β (SBDβ), that is known to be on recognition site of SARsCOv-2 spike show average binding energy −6.25 ± 1.10 and −5.520 ± 0.12 kcal/mol to with thymoquinone (*Elfiky, 2020*). Therefore it is probably used as an anti-covid-19 drug for the patient having pre-existing medical condition or stress. Green chiretta or Kalmegh (*Andrographis paniculata*) is an excellent medicinal plant having anti-HIV, anti-pathogenic, anti-viral, and immunoregulatory activities. Its bioactive constituent andrographolide has been reported to have immunoregulatory activities, plays a significant role as a modulator of altered immune responses. It can regulate both classical and alternative activation of macrophages, and specific antibody production as well as antigen-specific producing splenocytes. The insilico study (*Enmozhi et al., 2020*) conducted to evaluate the potential of Andrographolide against covid-19 infection showed high binding energy, better confirmation, bioactivity, and pharmacodynamics property against main protease (Mpro) protein of SARS-CoV-2. To study the potential of andrographolide (AGP1), 14-deoxy 11,12-didehydro andrographolide (AGP2), neoandrographolide (AGP3), and 14-deoxy andrographolide (AGP4), molecular dynamic and simulation study were performed against the central targets of virus namely non-structural proteins (3 L main protease (3CLpro), Papain-like proteinase (PLpro) and

RNA-directed RNA polymerase (RdRp)) and a structural protein spike protein (S). The finding suggests the effectiveness and potential of AGP3 analog against covid-19 infection (*Murugan, Pandian & Jeyakanthan, 2020*). An integrated study on liquid chromatography-tandem mass spectrometry (LC-MS/MS) metabolomics and network pharmacology on kalmegh suggested its active role as immune-protective and antiviral activity as well as its involvement in various pathway such as toll-like receptor pathway, PI3/AKT pathway, and MAP kinase pathways against SARs CoV-2 (*Banerjee et al., 2020*). Thus, this phytochemical can inhibit viral propagation and can act as an excellent antiviral drug. Therefore, the obtained compounds if synthesized and tested on an animal model will hold promise in the treatment of the SARS CoV-2 virus.

## CONCLUSION

The study concludes the high potential of Cinnamic acid, Thymoquinone, and Andrographolide against SARS-CoV-2 ACE2 receptor protein. The molecular docking & molecular dynamics simulation study reveals the better binding affinity, stability, and structural conformation at the binding site of these compounds against the ACE2 receptor. Therefore, the above finding will help to design a novel potent therapeutic inhibitor against SARS-CoV-2.

## ACKNOWLEDGEMENTS

We gratefully acknowledge Mr. Vinod Devaraj, Application Scientist, Schrodinger, Bangalore for his support provided in the molecular dynamics simulation study. We owe deep gratitude to the C.E.O, Biotech Park, Lucknow for their encouragement & support in the study. The support provided by Bioinformatics tools, software & databases in the work is also gratefully acknowledged.

### Funding

Prahlad Kishore Seth was awarded a Nasi Senior Scientist Platinum Jubilee fellowship from the National Academy of Sciences India. The funders had no role in study design, data collection and analysis, decision to publish, or preparation of the manuscript

### Grant Disclosures

The following grant information was disclosed by the authors:
National Academy of Sciences India.

### Competing Interests

The authors declare that they have no competing interests.

### Author Contributions

- Neha Srivastava conceived and designed the experiments, performed the experiments, analyzed the data, prepared figures and/or tables, authored or reviewed drafts of the paper, and approved the final draft.

- Prekshi Garg conceived and designed the experiments, performed the experiments, prepared figures and/or tables, and approved the final draft.
- Prachi Srivastava conceived and designed the experiments, authored or reviewed drafts of the paper, and approved the final draft.
- Prahlad Kishore Seth conceived and designed the experiments, analyzed the data, authored or reviewed drafts of the paper, and approved the final draft.

## Data Availability

Raw data is available in the Supplemental Files.

ACE2 human receptor file is available from NCBI having Accession ID Q9BYF1.2.

## Supplemental Information

Supplemental information for this article can be found online at http://dx.doi.org/10.7717/peerj.11171#supplemental-information.

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
