# Peer review of "A molecular dynamics simulation study of the ACE2 receptor with screened natural inhibitors to identify novel drug candidate against COVID-19"

_PeerJ, doi:10.7717/peerj.11171_

## Round 0.1 · original submission · Major Revisions

We got two reviews suggesting reject the manuscript in its current form. I think the presentation of the material should be improved. Please use the detailed comments from the reviewers. I encourage revise the text, justify the selection of the natural compounds (traditional medicine), and extend the computing. Please note the PeerJ journal policy on molecular docking calculations below.

The topic for coronavirus drug design is really important, but it is very competitive field. Please cite recent literature and approaches.

You are welcome to resubmit the manuscript to PeerJ after revision.

PeerJ policy is that submissions that screen molecules based on network pharmacology or molecular docking calculations must validate the candidate molecules either experimentally, or through molecular dynamics simulations. If the molecular dynamics route is selected, then duplicated simulations (at least 50 ns each) should be conducted on the best ligands (https://peerj.com/about/policies-and-procedures/#discipline-standards).

Reviewer 1 ·

Basic reporting

In my opinion, this manuscript has serious issues and cannot be published in its current form:
1) The authors do not provide any rationale under the necessity to inhibit the ACE2 receptor to fight COVID-19. Are ACE2 receptor inhibitors safe? Why inhibition of ACE2 is preferred in comparison with inhibition of SARS-CoV-2? It is not clear from the text of the manuscript.

2) The authors prepared the models of protein (ACE2 receptor), then they prepared the molecular docking and molecular dynamics of the complex protein-inhibitor. So if there is a low resolution of the model, therefore, the results of molecular docking and molecular dynamics are not characterized by high accuracy. I think the better strategy is, to use ligand-based methods, or at least to try to obtain the training set of known ACE2 receptors inhibitors and to make a conclusion whether the ligand-based drug design is suitable in this task.

3) The paper has many typos and unclear phrases:

"Coronavirus contains a genome of approximately 30kb and consists of a 5' cap structure and 3' poly(A) tail that allows it to act as an mRNA for the translation of the
replicase polyproteins. At the beginning of each structural or accessory gene, there is the
presence of transcriptional regulatory sequences (TRSs) which are essential for the expression of each of these genes." Is this phase related to the general idea of the text?


"The available 2D & 3D structures of compounds were retrieved from the PubChem database (Ref)" Is it correct reference here?

"(Palmitoyl Oleoyl Phasphatidyl chlorine" - there is a typo
etc.

Experimental design

1) The authors do not provide any rationale under the necessity to inhibit the ACE2 receptor to fight COVID-19. Are ACE2 receptor inhibitors safe? Why inhibition of ACE2 is preferred in comparison with inhibition of SARS-CoV-2? It is not clear from the text of the manuscript.

2) The authors prepared the models of protein (ACE2 receptor), then they prepared the molecular docking and molecular dynamics of the complex protein-inhibitor. So if there is a low resolution of the model, therefore, the results of molecular docking and molecular dynamics are not characterized by high accuracy. I think the better strategy is, to use ligand-based methods, or at least to try to obtain the training set of known ACE2 receptors inhibitors and to make a conclusion whether the ligand-based drug design is suitable in this task.

3) It is not clear why the authors chose Autodock 1.5.6 for molecular docking.

Validity of the findings

'no comment', please see "1. Basic reporting" field

Additional comments

Dear authors,
you have presented the results regarding the search for the potential ACE2 receptor inhibitors as the potential compounds for treatment COVID-19. Despite the fact, that in general, the idea is interesting, I'd recommend you to provide the answers to the comments given in the "1. Basic reporting" section. In particular, the main question is why the authors decided to model the ACE2 receptor and to find its inhibitors. What are the advantages of such a method? Some more comments are provided in the "1. Basic reporting" and "2. Experimental design" sections, I've copied and pasted it below for the convenience:

1) The authors do not provide any rationale under the necessity to inhibit the ACE2 receptor to fight COVID-19. Are ACE2 receptor inhibitors safe? Why inhibition of ACE2 is preferred in comparison with inhibition of SARS-CoV-2? It is not clear from the text of the manuscript.

2) The authors prepared the models of protein (ACE2 receptor), then they prepared the molecular docking and molecular dynamics of the complex protein-inhibitor. So if there is a low resolution of the model, therefore, the results of molecular docking and molecular dynamics are not characterized by high accuracy. I think the better strategy is, to use ligand-based methods, or at least to try to obtain the training set of known ACE2 receptors inhibitors and to make a conclusion whether the ligand-based drug design is suitable in this task.

3) It is not clear why the authors chose Autodock 1.5.6 for molecular docking.

4) The paper has many typos and unclear phrases:

"Coronavirus contains a genome of approximately 30kb and consists of a 5' cap structure and 3' poly(A) tail that allows it to act as an mRNA for the translation of the
replicase polyproteins. At the beginning of each structural or accessory gene, there is the
presence of transcriptional regulatory sequences (TRSs) which are essential for the expression of each of these genes." Is this phase related to the general idea of the text?


"The available 2D & 3D structures of compounds were retrieved from the PubChem database (Ref)" Is it correct reference here?

"(Palmitoyl Oleoyl Phasphatidyl chlorine" - there is a typo

I suggest that the authors either (i) choose another target for designing the potential drugs for the treatment of COVID-19 or (ii) provide the rationale of the usage of the ACE2 receptor as a target. Then, I suggest that the authors consider ligand-based drug design if there are inhibitors of the ACE2 receptor (or another chosen target) instead of 3D modeling methods, in case, when the 3D structure of the protein target is unknown.

Sincerely.

·

Basic reporting

The present study explores the potential of human Angiotensin-converting enzyme 2 (ACE2) receptor as the target for the designing of the drug against the deadly virus and marks an attempt to identify novel therapeutic inhibitor against novel SARS-CoV-2 using the Insilico drug discovery approach

1. Basic reporting
a. There is scope for improvement in Introduction, Discussion and Conclusion
b. Structure conforms to PeerJ quality; however, results, discussion and conclusion needs to be separated for better understanding of the international audience.
c. Figures are relevant and labelled. Authors need to highlight relevant figures in text (Section-Results/discussion wherever applicable) because they refer only Fig 3, Fig 4, and Fig 5 only in the text. Interestingly there are 11 figures in total in support of their findings.

Experimental design

Methods described in detail and information with possibility of replication

Validity of the findings

a. All underlying data have been provided; they are robust & controlled.
b. The conclusion is weak and very much limited supporting results.

Additional comments

a. Your introduction needs to be improved. I suggest that you add more relevant citation in support of your claims from lines 65-74; 78-79.
b. Highlight the text with suitable Figure or Table to familiarize or to engage audience/readers in lines from 75-130.
c. Fix typo errors thirty (Line 54); Ref? (Line 156); photochemical (line 252; phytochemical)
d. Line 39-In the present study; Line 55-proteins; 55-56 sentence needs to rephrase; Line 65-74-this statement needs to be supported with more references; Line 78-79 reference missing for claim; Line 131-141support with references; Line 188-Results. English language needs to be improvised

---

## Round 0.2 · Major Revisions

Thank you for the updates. Major remarks are fixed now. But there are still some remarks for data comparison and cross-validation. Please compare compounds reported using existing databases of chemical compounds such as ChEMBL, PubChem as noted by reviewer #1.
Compare your results with the prediction results of the web-services of biological activity prediction. There are recent publications on ACE2 modeling. Please add it at least to the discussion. Take more time for revision, if you need it. Waiting your revised manuscript.

Reviewer 1 ·

Basic reporting

I thank the authors for submitting to PeerJ.
I have some major comments that should be taken into account.
Please correct English (a few sentences are wordy and somehow unclear. For instance, the sentence: "AutoDock is an automated suite of protein-ligand docking tools that are designed to predict protein interactions with small molecules." can be replaced by the similar from the author's responce: "AutoDock is an automated suite of protein-ligand docking tools used to predict the protein interactions with small molecules such as drug molecule and substrate"., etc).

Experimental design

1. If the authors used a few compounds that can be potential inhibitors against ACE2, could they provide the explanation, why the authors used only those three compounds, that they mentioned?
2. I strongly recommend to add any results of analysis of the compounds using existing databases of chemical compounds such as ChEMBL, PubChem (for instance, exact search or similarity search). Also, it would be great, if the authors can complete their studies by the prediction results of the web-services of biological activity prediction and demonstate the association between the results of prediction and their findings.

Validity of the findings

I think that the authors should provide more data to prove their findings. For instancem they should add the results of the biological activity prediction or target affinity based on the publicly accessible web-services for prediction of either biological activity and/or target affinity.

Additional comments

I thank the authors for submitting to PeerJ.
I have some major comments that should be taken into account.
1. Please correct English (a few sentences are wordy and somehow unclear. For instance, the sentence: "AutoDock is an automated suite of protein-ligand docking tools that are designed to predict protein interactions with small molecules." can be replaced by the similar from the author's responce: "AutoDock is an automated suite of protein-ligand docking tools used to predict the protein interactions with small molecules such as drug molecule and substrate"., etc).
2. I think that you should provide more data to prove their findings. For instance, you should add the results of the biological activity prediction or target affinity based on the publicly accessible web-services for prediction of either biological activity and/or target affinity.
Kind regards.

·

Basic reporting

After through check with manuscript (track changes) it is evident that authors have amended necessary changes in key areas (Introduction, Methodology, Results, Discussion and Conclusion) with Tables and Figures. Furthermore, the literature is revised with references.

Experimental design

Methods described in detail and information with possibility of replication

Validity of the findings

All underlying data have been provided; they are robust & controlled.

Additional comments

Authors have amended the manuscript and addressed all the queries raised. The manuscript can now be accepted in present revised form.

---

## Round 0.3 · Minor Revisions

There are no remarks on science content, only on the text presentation. To follow the publication standards please check English again asking a professional agency or fluent English speaker.

After the update, we may accept it for publication without an additional reviewing round.

Reviewer 1 ·

Basic reporting

The authors implemented some changes regarding my comments.
But still, this manuscript needs English checking preferably with native speaker.

Experimental design

The authors implemented some changes regarding my comments.
But still, this manuscript needs English checking preferably with native speaker.

Validity of the findings

The authors implemented some changes regarding my comments.
But still, this manuscript needs English checking preferably with native speaker.

Additional comments

The authors implemented some changes regarding my comments.
But still, this manuscript needs English checking preferably with native speaker.

·

Basic reporting

Professional article structure, figures, tables are shared.

Experimental design

Methods described in detail and information with possibility of replication

Validity of the findings

All underlying data have been provided in revised manuscript; they are robust & controlled

Additional comments

The authors addressed all comments.

---

## Round 0.4 · accepted · Accept

Thanks for the text update. There are no more critical remarks. The topic is important itself. I believe the topic of novel drug candidate against COVID-19 using natural compounds could be extended in series of new studies.

Reviewer 1 ·

Basic reporting

Authors have corrected their manuscript to some extent.

Experimental design

Authors have corrected their manuscript to some extent.

Validity of the findings

Authors have corrected their manuscript to some extent.

Additional comments

Authors have corrected their manuscript to some extent.